# Malonylation of GAPDH is an inflammatory signal in macrophages

Silvia Galván-Peña[1,2], Richard G. Carroll[3], Carla Newman[4], Elizabeth C. Hinchy[5], Eva Palsson-McDermott[1], Elektra K. Robinson[6], Sergio Covarrubias [6], Alan Nadin[7], Andrew M. James[5], Moritz Haneklaus[1], Susan Carpenter[6], Vincent P. Kelly [1], Michael P. Murphy [5], Louise K. Modis[2] & Luke A. O'Neill[1,2]

Macrophages undergo metabolic changes during activation that are coupled to functional responses. The gram negative bacterial product lipopolysaccharide (LPS) is especially potent at driving metabolic reprogramming, enhancing glycolysis and altering the Krebs cycle. Here we describe a role for the citrate-derived metabolite malonyl-CoA in the effect of LPS in macrophages. Malonylation of a wide variety of proteins occurs in response to LPS. We focused on one of these, glyceraldehyde-3-phosphate dehydrogenase (GAPDH). In resting macrophages, GAPDH binds to and suppresses translation of several inflammatory mRNAs, including that encoding TNFα. Upon LPS stimulation, GAPDH undergoes malonylation on lysine 213, leading to its dissociation from TNFα mRNA, promoting translation. We therefore identify for the first time malonylation as a signal, regulating GAPDH mRNA binding to promote inflammation.

[1] School of Biochemistry and Immunology, Trinity Biomedical Science Institute, Trinity College, Dublin D2, Ireland. [2] Immunology Catalyst, GlaxoSmithKline, Stevenage SG1 2NY, UK. [3] Department of Molecular and Cellular Therapeutics, Royal College of Surgeons in Ireland, Dublin D2, Ireland. [4] In Vitro/In Vivo Translation, GlaxoSmithKline, Stevenage SG1 2NY, UK. [5] MRC Mitochondrial Biology Unit, University of Cambridge, Cambridge CB2 0XY, UK. [6] Department of Molecular Cell and Developmental Biology, UC Santa Cruz, Santa Cruz 95064 CA, USA. [7] NCE Molecular Tools Group, GlaxoSmithKline, Stevenage SG1 2NY, UK. Correspondence and requests for materials should be addressed to L.A.O'N. (email: laoneill@tcd.ie)

Post-translational modifications (PTMs) are key to expanding the functional diversity of proteins and have an important impact on protein function in health and disease[1]. Malonylation is a recently identified, evolutionarily conserved modification[2,3], wherein malonyl-CoA is used as a substrate to add a malonyl group to the amino acid lysine[2,4,5], changing its charge from +1 to −1. This change is predicted to disrupt electrostatic interactions with other amino acids and alter protein conformation and binding to targets[2]. Malonylation remains a poorly understood modification, with very few studies having investigated its functional impact. It has been shown to be present across various metabolic pathways, including fatty acid synthesis and oxidation[5,6], mitochondrial respiration[5] and glycolysis[6,7], as well as being capable of modifying histones[8]. It has also been recently linked to angiogenesis in endothelial cells by modifying mTOR complex 1 (mTORC1) kinase activity[9]. However, no further physiological function has yet been attributed to malonylation.

Several studies have recently highlighted the role of metabolic reprogramming in determining the function of immune cells (reviewed in ref. [10]). Macrophages have been a particular focus in this regard. These front line cells of innate immunity, inflammation, and tissue repair[11], display different metabolic features depending on their function. Pro-inflammatory macrophages, such as those activated by lipopolysaccharide (LPS), are highly glycolytic with a disrupted Krebs cycle[12,13]. Succinate has been shown to accumulate and drive production of reactive oxygen species, leading to the activation of hypoxia-inducible factor-1α and the induction of target genes, such as that encoding IL-1β[14,15]. Another Krebs cycle intermediate, citrate, accumulates, driving the production of inflammatory mediators, such as nitric oxide and prostaglandins[16], as well as the anti-inflammatory metabolite itaconate[17]. The role of the malonylation substrate and downstream metabolite of citrate, malonyl-CoA, is yet to be explored in immune cells and inflammation. Malonyl-CoA is synthesised in the cytosol from acetyl-CoA by acetyl-CoA carboxylase (ACC)[18] or in the mitochondria from malonate by ACSF3[4]. There are two different ACC isoforms; ACC1 is reported to be responsible for the production of malonyl-CoA in tissues with high levels of lipid synthesis and when knocked out in mice, it is embryonically lethal[19]. ACC2 is reported to be mostly expressed in oxidative tissues, where it can inhibit fatty acid oxidation via malonyl-CoA. ACC2 KO mice do not display embryonic lethality, but are resistant to obesity and diet-induced diabetes[20]. Interestingly, ACC1 has been recently shown to play a role in the differentiation of human CD4+ T cells into effector cells[21], as well as in the mechanism of defense of Th1 cells against *Mycobacterium tuberculosis* infection[22].

Here, we characterise malonylated proteins in an immune cell population. The malonylation response occurs in bone marrow-derived macrophages (BMDMs) following cell activation and relies on ACC1-dependent malonyl-CoA production. We found that malonylation of the glycolytic enzyme GAPDH in particular, has an impact on pro-inflammatory cytokine production, by modulating both its enzymatic activity and RNA-binding capacity. This novel finding reveals a hitherto unknown mechanism in LPS signalling that regulates the expression of central pro-inflammatory mediators, while further emphasising the importance of metabolic reprogramming in macrophage activation.

## Results

### MalonylCoA alters cytokine production in macrophages. To determine the role of malonyl-CoA in macrophages, we first set out to measure its production in LPS-activated BMDMs, and found it to be significantly increased following 24 h of LPS treatment (Fig. 1a). In order to identify the source of malonyl-CoA and be able to manipulate its levels, expression of the three existing malonyl-CoA-synthesising enzymes was analysed. We found the ACC1 isoform to be the highest expressed in BMDMs, followed by ACSF3, while no expression of the ACC2 isoform was detected. (Fig. 1b). We sought to compare our qPCR expression data with the existing RNAseq data from different immune cell types from the ImmGen consortium. The available RNAseq data from macrophages supports our results, with no ACC2 (*acacb*) expression detected and with ACC1 (*acaca*) being the highest expressed enzyme of the three (Supplementary Fig. 1). Furthermore, to our surprise, macrophages appear to have up to 10 times higher ACC1 expression than any other immune cell type (Supplementary Fig. 1a). Similarly, while ACC2 appears to be expressed in bone marrow immune stem cell precursors, it is not expressed in most immune cells, with the exception of some B cell populations, CD8+ T cells and FoxP3+ Treg cells (Supplementary Fig. 1b).

We sought to manipulate malonyl-CoA levels in BMDMs by knocking down ACC1 or ACSF3 using two independent siRNAs. ~50% knockdown (KD) of each enzyme was obtained as measured by gene expression and confirmed at the protein level in the case of ACC1 (Fig. 1c, d). ACC1 and ACSF3 KD resulted in a 30–50% reduction in basal malonyl-CoA and a 70% reduction in the LPS-elevated malonyl-CoA (Fig. 1e). Under these conditions, we evaluated cytokine production in LPS-treated BMDMs in order to assess what the functional consequences of reduced malonyl-CoA might be. We found that both ACC1 and ACSF3 KD decreased production of the pro-inflammatory cytokine IL6 (Supplementary Fig. 2a, 2b), whilst simultaneously boosting the production of the anti-inflammatory cytokine IL10 (Supplementary Fig. 2c, d). TNFα production was also reduced but only in the ACC1 KD, and interestingly, the inhibitory effect was only observed at the protein level (Fig. 1f), as TNFα transcription was unaffected (Fig. 1g), both following 6 and 24 h LPS treatment (Supplementary Fig. 2e). Next, to confirm that the distinct effects on TNFα were malonyl-CoA dependent, we sought to increase the metabolite's levels within the cells by treating them with malonyl-CoA. Treatment of BMDMs with malonyl-CoA resulted in an increase in intracellular malonyl-CoA levels (Supplementary Fig. 2g). Pre-treatment of ACC1 KD BMDMs with malonyl-CoA was able to recover the decrease in TNFα (Fig. 1h), thus confirming that malonyl-CoA can act as a modulator of cytokine production in macrophages.

### Activation of macrophages results in protein malonylation. Given the increase in malonyl-CoA and the observed functional consequences of inhibiting its production, we next investigated protein malonylation in response to LPS, as this would be a likely outcome of malonyl-CoA accumulation. As shown in Fig. 2a, LPS increased overall lysine malonylation (mal-K) on multiple proteins. Having tested the specificity of the antibody through peptide competition (Supplementary Fig. 3), we proceeded to validate this increase and identify the substrates of the LPS-induced malonylation through mass spectrometry. Malonylated proteins from untreated and LPS-treated macrophages were affinity purified and analysed via LC–MS/MS (Supplementary Fig. 4a, Supplementary Data 1). More than 80% of proteins identified were found to have a maximum of two malonylated sites on each protein (Supplementary Fig. 4b). Label-free quantification[23] was applied to the 412 quantifiable proteins and 843 quantifiable sites were identified on a range of proteins. Two hundred and eighty sites were found to be upregulated by LPS in total. Seventy eight of these were only found in LPS-treated samples, while 202 sites were found to be upregulated by more than 1.5 fold by LPS

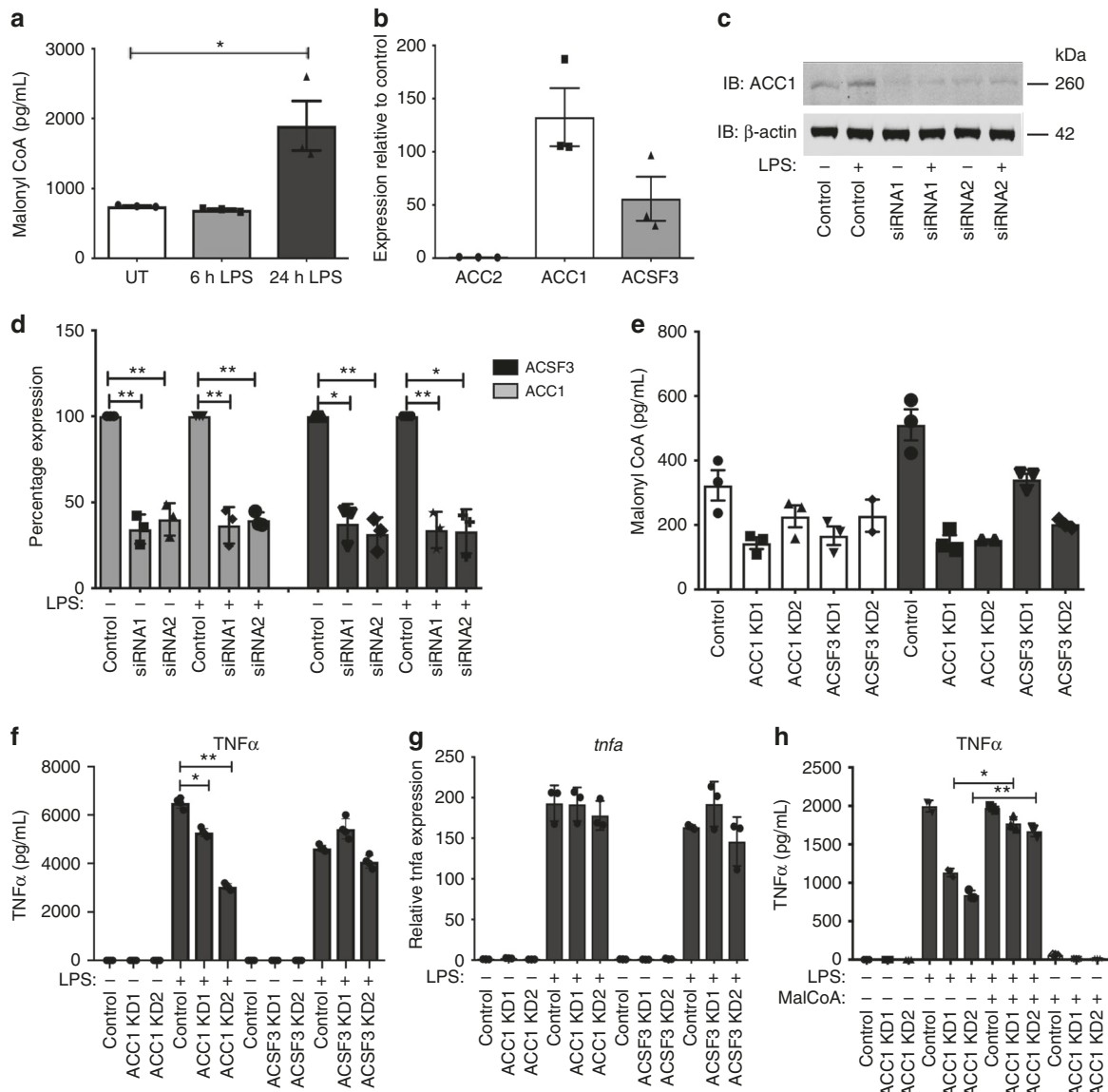

**Fig. 1** Activation of macrophages increases malonyl-CoA levels and is needed for pro-inflammatory cytokine production. **a** Malonyl-CoA levels measured in untreated and LPS-treated (100 ng/mL) BMDM lysates using a malonyl-CoA ELISA. Mean + SEM, $n = 3$. **b** ACC1, ACC2 and ACSF3 mRNA expression measured via qPCR in BMDMs. Mean + SEM, $n = 3$. **c** ACC1 siRNA (48 h, 10 nM) knockdown (KD) measured by western blotting in untreated and LPS-treated (100 ng/mL, 24 h) BMDMs. β-actin used as a control. Representative of three independent experiments. **d** ACC1 and ACSF3 siRNA (48 h, 10 nM) KD levels relative to control siRNA, measured via qPCR in untreated and LPS-treated (100 ng/mL, 24 h) BMDMs. Mean + SEM, $n = 4$. **e** Malonyl-CoA levels measured in ACC1 and ACSF3 KD BMDM lysates using a malonyl-CoA ELISA. Mean + SD, representative of four independent experiments. **f** TNFα protein measured by ELISA (mean + SEM, $n = 3$) and **g** TNFα mRNA expression in 100 ng/mL LPS-treated (100 ng/mL, 6 h) ACC1 and ACSF3 KD BMDMs (mean + SD, representative of three independent experiments). **h** TNFα protein measured by ELISA in ACC1 KD BMDMs pre-treated with malonylCoA (1 mM, 2 h) followed by LPS (100 ng/mL, 6 h) (mean + SD, representative of three independent experiments). Unpaired $t$-test, $*P < 0.05$; $**P < 0.005$

compared to untreated, 98 of which were found to be statistically significant (Supplementary Data 1, Supplementary Fig. 4b). The majority of LPS-induced malonylated proteins are cytosolic, with 31% nuclear proteins, and only 7% mitochondrial proteins (Supplementary Fig. 5a). Unlike previous studies, where malonylation has mainly been shown to influence metabolic enzymes[5–7], a functionally diverse set of proteins was shown to undergo malonylation in response to LPS, including proteins involved in RNA regulation, signal transduction, immune signalling, and glycolysis (Fig. 2b, Supplementary Fig. 5b). One of the proteins identified in the analysis was glyceraldehyde-3-phosphate dehydrogenase (GAPDH). This is a critical enzyme in glycolysis but can also bind RNAs directly[24,25], and regulate

translation within the interferon-γ-activated inhibitor of translation (GAIT) complex[26]. Given the role of glycolysis in LPS action in macrophages, as well as its RNA-binding capacity, we chose GAPDH as a good candidate to investigate the role of LPS-induced malonylation in macrophage activation.

**LPS induces malonylation of GAPDH at lysine 213**. We immunoprecipitated GAPDH from BMDM lysates and showed that in agreement with the mass spectrometry results, LPS strongly increased its malonylation by western blotting with an anti-mal-K antibody (Fig. 2c, lower panel, compare lane 5 to lanes 6–8). GAPDH LPS-induced malonylation was observed no earlier

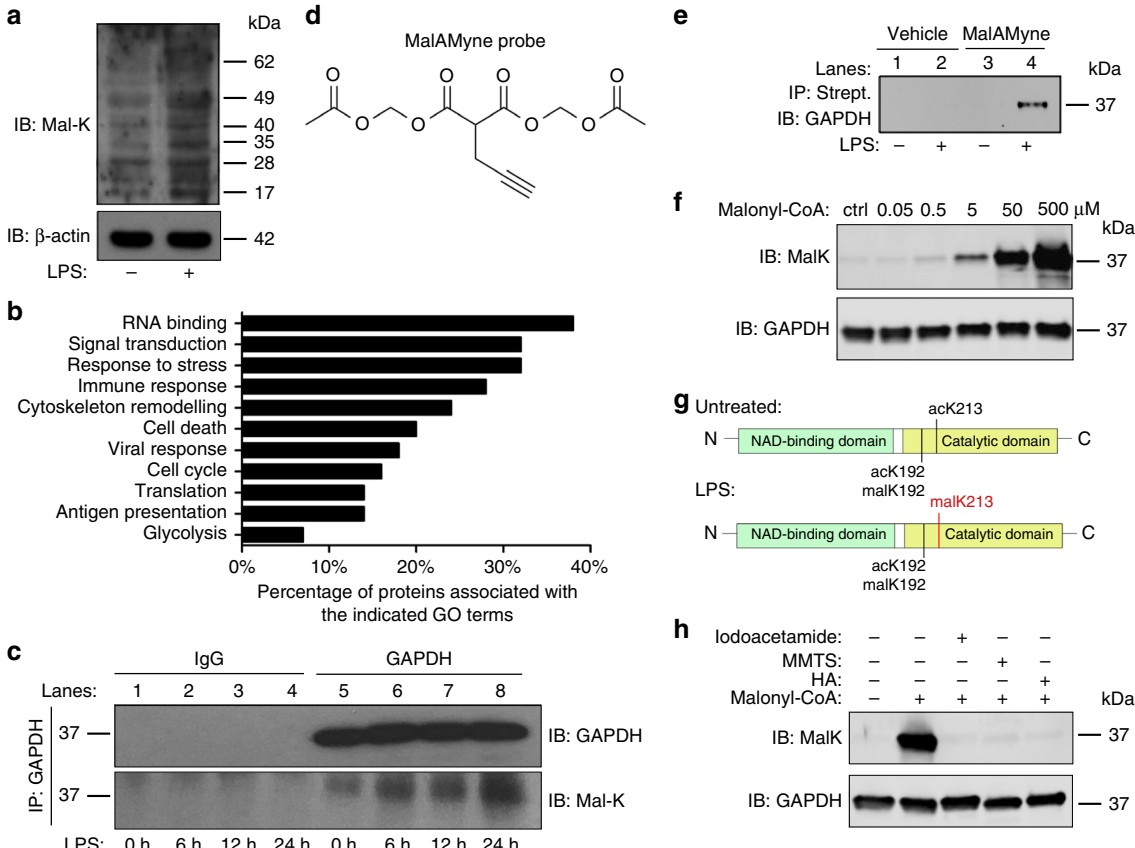

**Fig. 2** Activation of macrophages with LPS increases protein malonylation, with GAPDH as a substrate. **a** Western blot analysis of lysine malonylation (mal-K) in lysates from BMDMs treated with LPS (100 ng/mL) for 24 h. **b** Most enriched functions associated with LPS-induced malonylated proteins (FDR < 0.05). **c** Immunoprecipitated GAPDH from untreated and LPS-treated (100 ng/mL, 6, 12 and 24 h) BMDMs and samples probed with an anti-malK antibody (lower panel). GAPDH expression in the immunoprecipitated (upper panel) samples was also examined. **d** Chemical formula of the MalAMyne probe. **e** Untreated and LPS-treated (100 ng/mL, 24 h) BMDMs were labelled with MalAMyne (10 μM) or vehicle control. Copper-catalysed click chemistry was performed on the lysates using biotin, followed by immunoprecipitation using streptavidin (strept.). Samples were probed for GAPDH via western blotting. **f** Purified GAPDH (100 μg/mL) was incubated in the presence of TCEP and malonyl-CoA (or buffer as control) for 1 h at 37 °C, pH 7.5. K-malonylation was assessed by western blotting. **g** Identification of malonylated sites in immunoprecipitated trypsin-digested GAPDH peptides from untreated and LPS-treated BMDMs ($10^6$ cells). Peptides were analysed via LC–MS. **h** Purified GAPDH was preincubated with TCEP and 25 mM iodoacetamide, 80 mM methyl methanethiosulfonate (MMTS), 5 μM heptelidic acid (HA) or buffer for 30 min at 37 °C, followed by 500 μM malonyl-CoA. K-malonylation was assessed by western blotting. All data shown are representative of three independent experiments

than 6 h and seemed to gradually increase, with a much more noticeable increase at 24 h. Furthermore, we also used the malonylation chemical probe, malAMyne (Fig. 2d), and methodology by X. Bao et al.[27], to label malonylated lysines in untreated and LPS-treated BMDMs. This additional method further confirmed that LPS strongly induced malonylation of GAPDH (Fig. 2e, compare lane 4 to lane 3). In addition, we found that GAPDH in vitro is highly sensitive to malonylation by low concentrations of malonyl-CoA (Fig. 2f).

Seven malonylated sites had been originally identified on GAPDH (Supplementary Data 1). We next sought to determine which of these would be the most abundant and physiologically relevant by immunoprecipitating endogenous GAPDH from BMDMs and identifying the PTMs present by mass spectrometry. In this manner, only two lysines were identified as undergoing malonylation (Fig. 2g, Supplementary Fig. 6). Lysine 192 (K192) was malonylated in GAPDH in resting macrophages, but importantly lysine 213 (K213) underwent malonylation following macrophage activation with LPS (Supplementary Data 2). Interestingly, the same lysines were also found to be acetylated. The residue K213 is highly conserved (Supplementary Fig. 7a). It is present both, in the catalytic domain in close proximity to the

enzyme's active site cysteine (Supplementary Fig. 7b), as well as within the dimerisation region, in close proximity to the key threonine residue previously linked to RNA-binding and GAPDH dimerisation (Supplementary Fig. 7c)[28].

We next aimed to address the potential mechanism of malonylation. No malonyltransferase enzymes have yet been identified, although both protein succinylation and malonylation have been shown to be removed by the same enzyme[29]. We first examined whether the recently identified succinyl transferase KAT2A[30] could be a malonyl-transferase for GAPDH. However, we did not find KAT2A to interact with GAPDH (Supplementary Data 3). Following LPS treatment at the time of GAPDH malonylation, GAPDH was found to interact with the acetyl-transferase p300, which has also recently been identified as a crotonyl-transferase[31], and might therefore be a good GAPDH malonyl-transferase candidate[32,33]. On the other hand, it has been previously shown that reactive cysteine residues can catalyse the transfer of acyl groups from CoA onto proximal lysine residues within the same protein[34]. As the active site cysteine residue of GAPDH is within 18.4 Å of K213 (Supplementary Fig. 7b), we hypothesised that K213 could be particularly susceptible to malonylation due to its proximity. We found that

rapid GAPDH in vitro malonylation could be prevented by the alkylating agents iodoacetamide and methylmethanethiosulfonate, and more importantly, by the GAPDH inhibitor heptelidic acid (HA), which selectively alkylates the active site cysteine of GAPDH[35] (Fig. 2h). These findings suggest that K213 is particularly sensitive to malonylation in response to the elevation of cytosolic malonyl-CoA upon LPS treatment, and might indicate that GAPDH can perhaps catalyse its own malonylation under certain conditions.

**GAPDH is needed for cytokine production in macrophages.** We next explored the role of GAPDH malonylation on K213 during LPS activation in detail, first exploring the effect of malonylation on GAPDH activity. Consistent with previous reports showing that pro-inflammatory macrophages have increased glycolytic flux[12,36], the activity of GAPDH in LPS-treated macrophages was increased after 24 h (Fig. 3a). No changes in protein expression were detected (Supplementary Fig. 8a), however, we had previously shown malonylation of GAPDH to be highest at this time. Furthermore, inhibiting GAPDH enzymatic activity with HA[37] (Supplementary Fig. 8b), reduced transcription of the pro-inflammatory cytokine pro-IL1β, to the same extent as another glycolytic inhibitor, 2-deoxyglucose (2-DG), as shown previously[14] (Supplementary Fig. 8c). The induction of pro-IL1β protein was also inhibited by HA (Supplementary Fig. 8d). IL6 mRNA and protein (Supplementary Fig. 8e, f) were also inhibited by HA. Intriguingly, a different result was obtained with TNFα. A clear inhibitory effect was observed at the protein level with HA but not 2-DG (Fig. 3b). HA did not inhibit the induction of TNFα mRNA however (Fig. 3c), while 2DG boosted TNFα mRNA

production. We further explored the role of GAPDH in cytokine production using siRNA to KD GAPDH in BMDMs (Fig. 3d, Supplementary Fig. 9a). Consistent with the inhibitory effect of HA, KD of GAPDH with two independent siRNAs resulted in a decrease in pro-IL1β (Fig. 3d, Supplementary Fig. 9b) and IL6 production (Supplementary Fig. 9c, d). GAPDH KD had however, the opposite effect on TNFα, the production of which was boosted (Fig. 3e), while transcript levels remained unchanged (Fig. 3f). These findings confirm a glycolysis-dependent regulation of pro-IL1β and IL6, as shown by both glycolytic inhibitors and GAPDH KD having the same effect. On the other hand, a glycolysis-independent, post-transcriptional mechanism of regulation of TNFα production mediated by GAPDH is indicated by the lack of effect of 2DG on TNFα. The inhibitory effect of HA on TNFα protein production was intriguing, indicating that targeting of GAPDH by HA would perhaps boost mRNA binding by GAPDH, thereby repressing its post-transcriptional processing. A possible mechanism is HA preventing GAPDH malonylation, implying that malonylation could be a signal to dissociate GAPDH from TNFα mRNA, allowing for its post-transcriptional processing. The differing effects of 2DG, HA and GAPDH KD are depicted in Supplementary Fig. 9e. By attenuating glycolysis, induction of IL1β is blocked, as expected. Glycolysis per se has no role in TNFα production, however, GAPDH represses TNFα, possibly because of RNA binding blocking translation. LPS relieves this repression by causing the dissociation of GAPDH from the mRNA, which requires GAPDH malonylation.

**GAPDH regulates TNFα production via RNA-binding.** To test the possibility that GAPDH may be post-transcriptionally

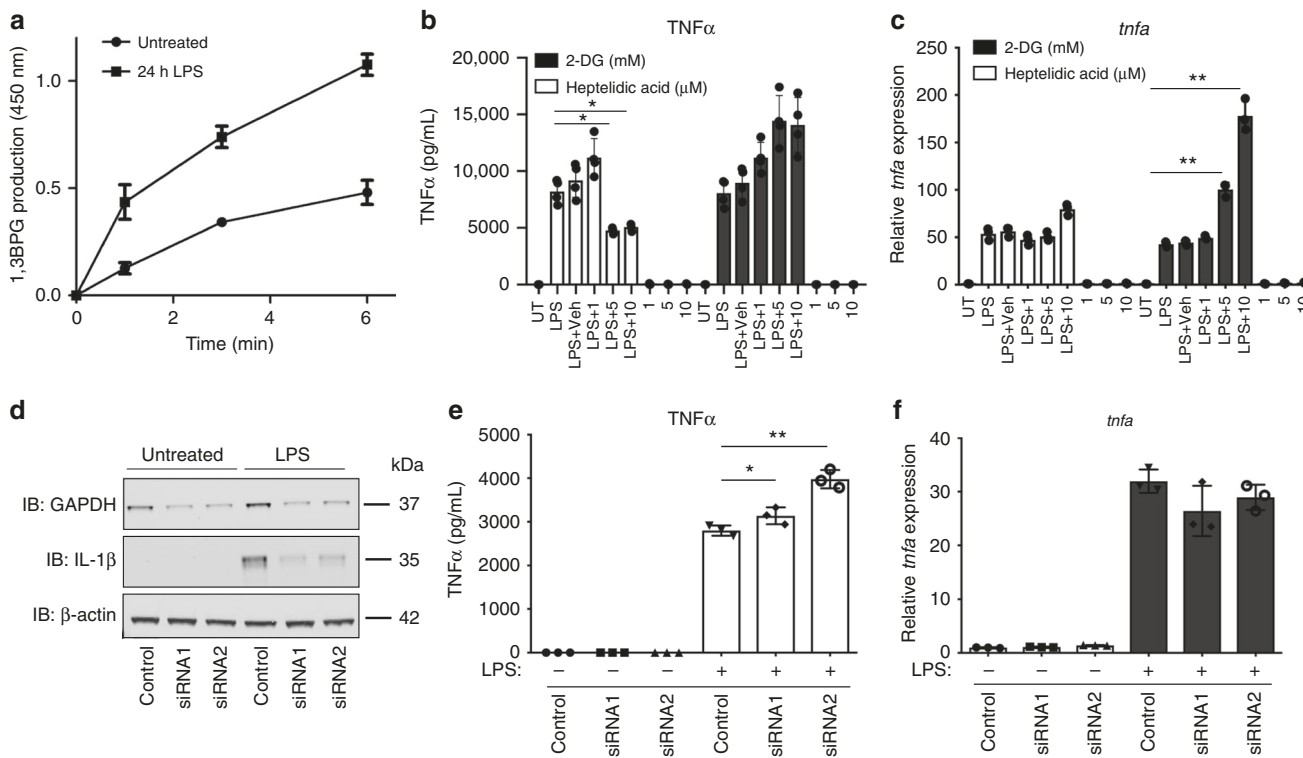

**Fig. 3** GAPDH controls TNFα production. **a** Untreated, 6 and 24 h LPS-treated (100 ng/mL) BMDM lysates were assayed for GAPDH enzymatic activity by monitoring product production (1,3 bisphosphoglycerate, 1,3BPG) over time. Representative of three independent experiments. **b** TNFα protein measured by MSD (mean + SEM, $n = 4$) and **c** TNFα mRNA expression in 100 ng/mL LPS-treated BMDMs (6 h), pre-treated with HA or 2-DG. **d** GAPDH siRNA (72 h, 10 nM) KD in BMDMs, measured by western blotting. Pro-IL1β levels were also measured, and β-actin used as a control. Representative of three independent experiments. **e** TNFα measured in GAPDH KD BMDMs, treated with LPS (24 h, 100 ng/mL, mean + SEM, $n = 3$). **f** TNFα mRNA expression measured by qPCR in LPS-treated GAPDH KD BMDMs. Unpaired t-test, *$p < 0.05$; **$p < 0.01$

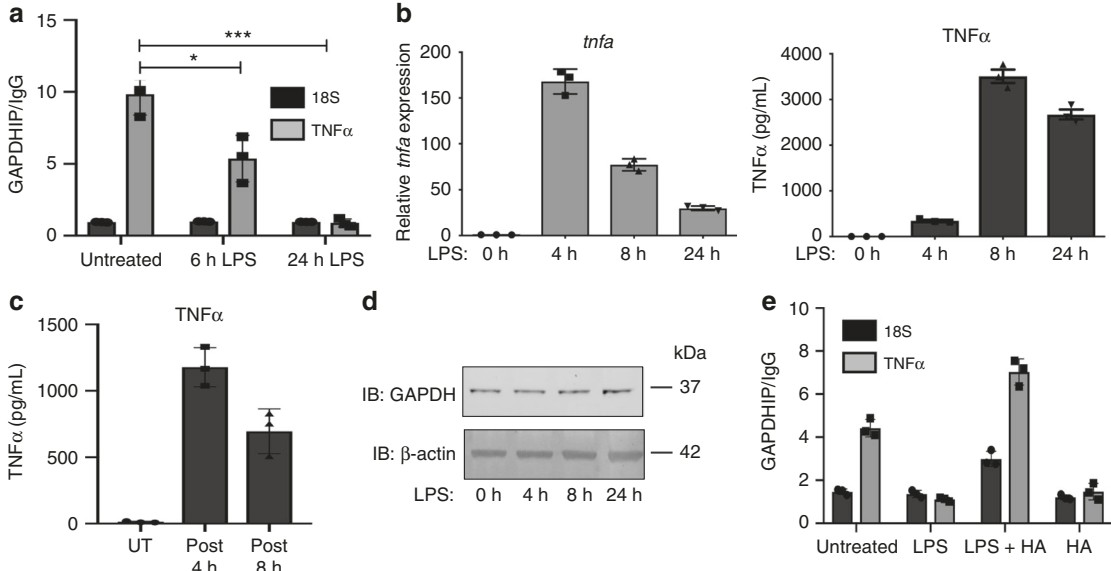

**Fig. 4** GAPDH regulates TNFα post-transcriptional regulation. **a** GAPDH was immunoprecipitated from untreated and LPS-treated (6 and 24 h, 100 ng/mL) BMDMs, and TNFα mRNA presence in IP relative to IgG control assessed via qPCR. (mean + SEM, n = 3). **b** TNFα mRNA expression measured by qPCR and TNFα protein measured by ELISA in BMDMs treated with LPS (100 ng/mL) over time. **c** TNFα protein measured by ELISA in BMDMs treated with LPS (100 ng/mL). Cells were washed post 4 h and post 8 h LPS treatment, and supernatants replaced (without LPS) and harvested 20 and 16 h later, respectively (mean + SEM, n = 3). **d** GAPDH expression analysed by western blotting in LPS-treated BMDMs. **e** LPS-treated (100 ng/mL, 24 h) BMDMs were pre-treated with HA (10 μM) and GAPDH immunoprecipitated. Bound TNFα mRNA was assessed by qPCR. Data shown are representative of three independent experiments. Unpaired t-test, *p < 0.05; ***p < 0.01

regulating TNFα production directly, we turned our attention to RNA-binding by GAPDH. GAPDH has been previously reported to bind mRNA transcripts containing AU-rich elements[38,39], which are present in TNFα[40,41], as well as IFNγ mRNAs[42]. It has also been shown to be part of the GAIT complex, which mediates translational repression of GAIT element-containing transcripts[26]. We first immunoprecipitated GAPDH from BMDMs and examined bound RNAs by qPCR. GAPDH in resting macrophages was found to bind the RNA of TNFα (Fig. 4a). Following 6 h activation with LPS, there was a significant reduction in binding to TNFα RNA, and following 24 h, GAPDH binding could no longer be detected (Fig. 4a). Similarly, in resting macrophages GAPDH bound to mRNA for the GAIT-element-containing death-associated protein kinase-1 (DAPK1) and dissociated following LPS treatment (Supplementary Fig. 10a). Enhanced translation appeared likely to be a key mechanism for TNFα and DAPK1 production, since there was a clear discrepancy between mRNA and protein induction for both genes. Both TNFα and DAPK1 mRNA levels were low when protein was high (Fig. 4b, Supplementary Fig. 10b). The 3′-untranslated region (UTR) of both genes was also repressive as they both inhibited reporter expression from luciferase vectors expressing DAPK1 and TNFα 3′-UTRs (Supplementary Fig. 10c). To validate that BMDMs are actively translating and secreting TNFα after the initial LPS stimulus, cells were treated with LPS for 4 and 8 h. The LPS-containing media was then removed, replaced with fresh media not containing any LPS, and the supernatants were harvested 24 h after the original LPS stimulus. Despite the cells having been without LPS for 20 and 16 h, respectively, they were still able to synthesise and secrete TNFα (Fig. 4c), indicating that the increase in TNFα production over time depicted in Fig. 4b is not due to continuous TNF α transcription in response to direct LPS sensing. Furthermore, we also investigated the translation efficiency of Tnfα in primary BMDMs pre-LPS and post-LPS stimulation by evaluating relative polysome enrichment (Supplementary Fig. 11). RNA was isolated from 80S (monosomes), low-molecular-weight and high-molecular-weight fractions. Gapdh

mRNA, which is not regulated by LPS treatment, was studied as a high polysome control, while Neat1, a long noncoding RNA (lncRNA), was studied as an 80S polysome control. Under these settings we found Tnfα mRNA to be enriched in the low polysome fraction with a marked increase in enrichment in the LPS-treated samples (Supplementary Fig. 11d), indicating that there is active TNFα translation in LPS-treated macrophages at the time of GAPDH malonylation. In addition, we found that following LPS treatment, GAPDH can be found directly interacting with various ribosomal components (Supplementary Fig. 12a, Supplementary Data 3), which would indicate GAPDH is able to directly transfer the RNA onto the translation machinery to enable translation.

Overall, we have observed TNFα protein production to be enhanced in BMDMs when GAPDH was knocked down, GAPDH dissociating from the TNFα mRNA in response to LPS, as well as interacting with the translation machinery at the time of active TNFα translation. All these results taken together support a role for GAPDH as an RNA-binding protein responsible for suppressing translation in resting macrophages, and dissociating from target mRNAs in response to LPS.

We next sought to determine whether malonylation of GAPDH might be involved in GAPDH dissociation from target mRNAs. GAPDH protein expression is not affected by LPS treatment (Fig. 4d), suggesting that PTMs might indeed be the mechanism by which LPS could alter GAPDH activity. Treatment of BMDMs with HA prior to LPS, which would prevent GAPDH malonylation, resulted in a profound increase in binding of GAPDH to TNFα (Fig. 4e) and DAPK1 mRNA (Supplementary Fig. 12b), as shown by GAPDH RNA-immunoprecipitation. These results therefore might explain why HA can block TNFα production, as it will maintain GAPDH in its unmalonylated state on the TNFα 3′-UTR and suppress its translation.

**Malonylation of GAPDH controls the enzyme's activities.** Having identified GAPDH as a two-way regulator of cytokine

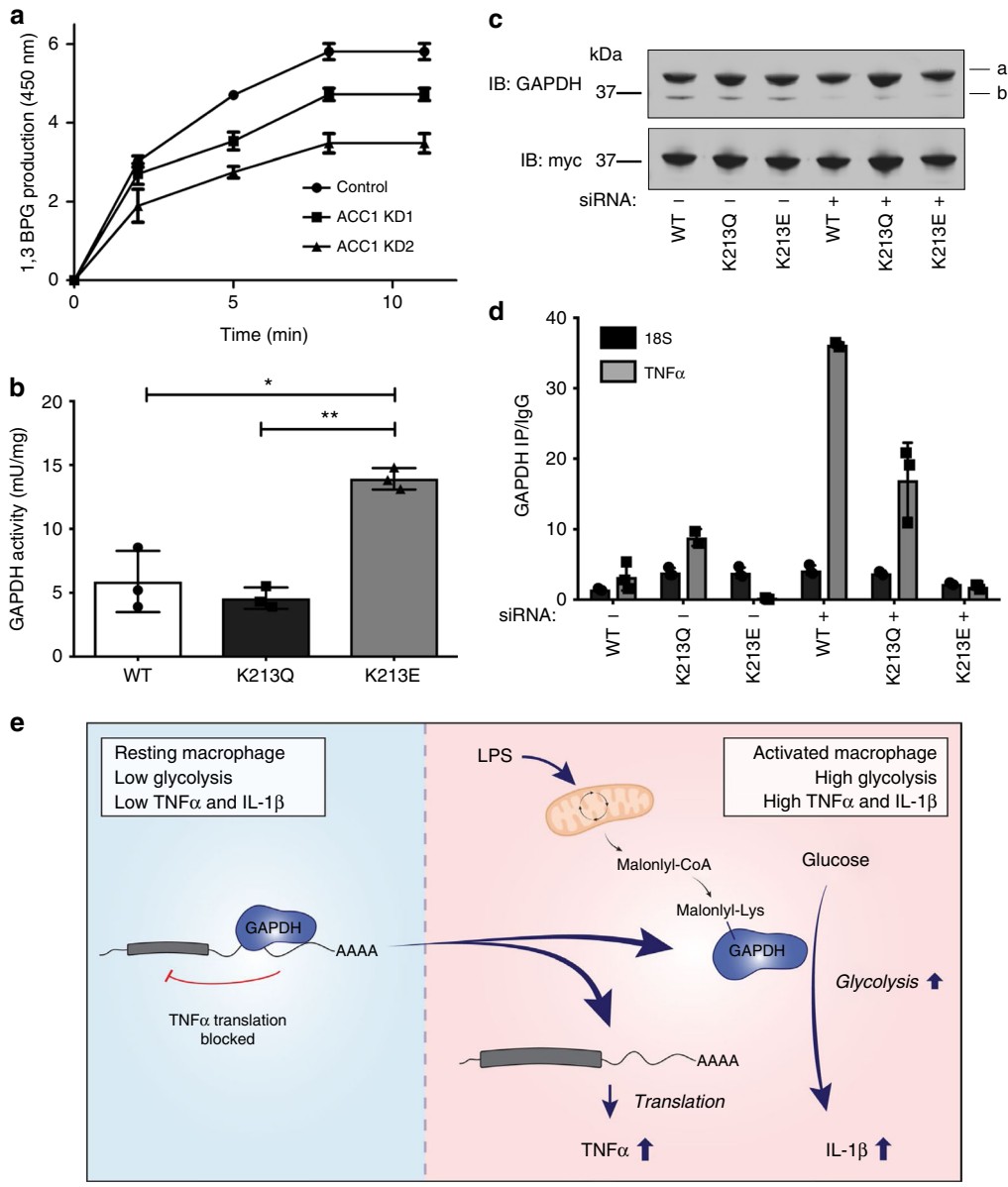

**Fig. 5** K213 malonylation regulates GAPDH enzymatic activity and RNA-binding. **a** GAPDH enzymatic activity measured in lysates from ACC1 and ACSF3 KD BMDMs (LPS treated, 100 ng/mL, 24 h; mean + SD). **b** WT, K213Q and K213E GAPDH mutants were overexpressed in HEK293T, affinity purified, and the enzymatic activity measured (mean + SEM, $n = 3$). **c** GAPDH was knockdown using siRNA (48 h, 10 nM) followed by overexpression of WT, K213Q and K213E GAPDH mutants and expression assessed via western blotting (a, overexpressed GAPDH; b, endogenous GAPDH). **d** GAPDH mutants were immunoprecipitated following 48 h overexpression and fixing in HEK293T cells. Bound TNFα mRNA in the IP relative to IgG was assessed via qPCR. Mean + SD, representative of three independent experiments. Unpaired t-test, *$p < 0.05$; **$p < 0.01$. **e** In resting macrophages, GAPDH binds to the 3′-UTR of various pro-inflammatory mediators, such as TNFα, and prevents their translation. Following activation of macrophages with LPS, there is an accumulation of citrate that can be converted into malonyl-CoA in an ACC1-dependent manner. Malonyl-CoA can then in turn mediate an increase in lysine malonylation. One of the substrates of this modification is GAPDH, which after undergoing malonylation, releases the bound RNAs which can now be translated. At the same time, GAPDH enzymatic activity increases, which allows for an increased glycolytic flux, required for the activated macrophage to carry out its pro-inflammatory functions

production in activated macrophages, we next set out to identify the mechanism behind GAPDH activities. Our data in Fig. 1e supported a role for GAPDH malonylation in TNFα production, since reduced malonyl-CoA levels in ACC1 KD cells also resulted in reduced TNFα production in response to LPS. GAPDH expression was unaffected in ACC1 KD and ACSF3 KD BMDMs (Supplementary Fig. 13a, b), however the activity of GAPDH was reduced in LPS-treated ACC1 KD cells (Fig. 5a). This implied that malonylation of GAPDH frees the enzyme from bound mRNA enabling it to enhance glycolytic flux. The same was not

the case in ACSF3 KD, where activity of GAPDH was unaffected (Supplementary Fig. 13c). ACSF3-derived malonyl-CoA and subsequent malonylation are confined to the mitochondria[4], while GAPDH is present in the cytosol, thus explaining the observed differences. Consistent with our results demonstrating that LPS both increased GAPDH enzymatic activity while decreasing GAPDH RNA-binding, these activities have been previously demonstrated to be mutually exclusive[38,39]. The evidence therefore indicates that ACC1-derived malonyl-CoA is needed for GAPDH malonylation, which in turn is needed for

TNFα production via dissociation from its 3′-UTR, as well as for its increased enzymatic activity.

To address the role of GAPDH K213 malonylation specifically in these two activities, a K213 glutamine mutant (K213Q) was generated. K213Q acts both as an acetylation mimic and as a control, being unable to undergo malonylation. We also generated a K213 glutamate (K213E) mutant, to act as a malonylation mimic[7]. We first determined the effect of these mutations on GAPDH enzymatic activity by affinity purifying them from transfected HEK293T cells (Supplementary Fig. 14a, b) and assaying them. The K213E mutant was found to have much higher enzymatic activity than both WT and K213Q (Fig. 5b). This supported the idea that malonylation would have the ability to positively affect enzymatic activity by suppressing its mRNA binding. We next investigated the RNA-binding capacity of the GAPDH mutants, overexpressing them in HEK293T cells in which endogenous GAPDH had been knocked down (Fig. 5c). In direct opposition to the enzymatic activity, the K213E could not bind TNFα mRNA, in contrast to the K213Q mutant and wild type (Fig. 5d). These results taken together indicate that malonylation of K213 increases GAPDH enzymatic activity, while blocking its binding to target mRNAs. This increased enzymatic activity and reduced RNA-binding within macrophages, enables the production of pro-inflammatory cytokines in response to LPS. GAPDH is needed for IL1β production because of the role of glycolysis in the process, and it must also dissociate from the TNFα mRNA in order for TNFα translation to occur. We therefore report a role for GAPDH malonylation on K213 in the translation of LPS-induced genes, notably that encoding TNFα. This scenario is depicted in Fig. 5e.

## Discussion

PTMs play an essential role in the regulation of protein activity and function across cellular systems, and despite their different functional impacts, the great majority of them share their origins in metabolic intermediates. Metabolic reprogramming in immune cells has repeatedly been shown to be key for immune function in recent years, and yet the connection between metabolic changes and PTMs remains a very poorly understood area. Our study reports the role of the post-translational modification malonylation in macrophage function. We have identified multiple proteins with a wide range of functions ranging from metabolism, to cell death, to immune responses, as undergoing malonylation following activation of cells with LPS. This effect is dependent on malonyl-CoA production by one of the two ACC isoforms, ACC1, which is highly expressed in macrophages. We propose a model whereby the previously reported accumulation of citrate in activated macrophages[13], exits the mitochondria and following conversion into acetyl-CoA, it is used to generate malonyl-CoA by ACC1 in the cytosol which in turn, can act as a substrate for malonylation of multiple substrates. LPS does not have any effect on ACC1 expression levels, indicating that the increase in malonyl-CoA production observed is likely the result of LPS modulating the enzyme's activity. Citrate has been shown to be able to allosterically regulate ACC activity directly[43], so the reported accumulation of citrate in activated macrophages might be the mechanism for the observed effects. Interestingly, we find no expression at all of the second ACC isoform, ACC2, an observation which is based on the currently available RNAseq data, and might not be restricted to just macrophages but to most immune cells. ACC1 is generally found in tissues where fatty acid synthesis is important, while ACC2 is associated with tissues with predominant oxidative metabolism. Their expression pattern thus fits with the existing literature showing fatty acid synthesis is

upregulated and needed for function in macrophages[44,45], dendritic cells[46] and T cells[21,22,47,48]. In addition, oxidative phosphorylation has been shown to be inhibited by nitric oxide[49], which is a hallmark of activated pro-inflammatory dendritic cells and macrophages[49,50], which might explain the absence in ACC2 expression.

While we have presented ACC1 as the source of malonyl-CoA for the malonylation reaction, the precise molecular details of the modification remain to be elucidated. The only known regulator of malonylation is sirtuin 5, which can act not only as a demalonylase, but can also remove similar acylations, such as succinylation and glutarylation[29,51]. It is usually the case that deacylases and acyl-transferases, as with sirtuin 5, can catalyse the removal or addition of closely similar acylations. Recently, two known acetyltransferases, KAT2A and p300, have been identified as the first succinyltransferase and crotonyltransferase, respectively[30,31]. It is possible that one of the existing acyl-transferases may be capable of catalysing the malonylation reaction as well. On the other hand, malonylation may be entirely dependent on the available pool of malonyl-CoA and affinity for different proteins. Given the increasing number of newly discovered metabolically derived modifications in the last decade, we predict the identification of enzymes and mechanisms behind these, to be an exciting area of research in the coming years.

We have identified the glycolytic enzyme GAPDH as one of the LPS-induced malonylated substrates in macrophages, undergoing malonylation of a specific lysine within its catalytic domain, lysine 213. While it has been reported numerous times that activated macrophages display increased glycolysis, studies into the role of GAPDH itself are lacking. The evidence we have presented here indicates that following macrophage activation, malonylation increases GAPDH enzymatic activity, enabling the production of pro-inflammatory cytokines, including IL1β and IL6. At the same time, we have shown that GAPDH can also bind RNAs, including that of TNFα, with this being the first report to our knowledge of malonylation affecting a protein function beyond enzymatic activity. GAPDH binds AU-rich elements in the 3′UTR of mRNA, an activity that is reported to have two possible outcomes; on the one hand, it may result in mRNA stabilisation, thus enabling translation, as is the case for GAPDH binding to colony stimulating factor 1 in ovarian cancer cells[52]. On the other hand, GAPDH binding may sequester the mRNA and prevent translation, as has been shown to be the case for IFNγ in T cells. Binding of GAPDH to TNFα mRNA in macrophages occurs as per the latter. In resting cells without a need for TNFα production, the mRNA is sequestered, thus preventing its translation. Following a few hours of LPS activation, if the cell finds itself still in need for TNFα production, the GAPDH-bound pool of TNFα mRNA is released and translated, enabling the amplification of the TNFα response, while at the same time freeing GAPDH to engage in glycolysis, resulting in a two-way wave of pro-inflammatory cytokine production. Interestingly, protein translation has been identified as one of the main biological functions influenced by malonylation in bacteria, where GAPDH K213 has also been identified as undergoing malonylation[53], indicating that this process is likely to be evolutionarily conserved.

In the present study, we have therefore identified malonylation as a novel mechanism by which macrophages can control the production of pro-inflammatory cytokines through GAPDH. A role for our identified model in various inflammatory settings is supported by recent reports. Administration of GAPDH in a mouse model of sepsis, has been shown to have anti-inflammatory effects by reducing TNFα[54], with our study providing a possible mechanism for this repressive effect. In mouse models of type 2 diabetes, GAPDH was shown to be

hypermalonylated, which includes increased malonylation of K213[6]. Targeting GAPDH through HA has also been shown to be beneficial in mouse models of breast cancer[55]. Furthermore, a recent study indicating that dimethylfumarate targets the active site cysteine in GAPDH and thereby elicits its effects as therapy in multiple sclerosis[56], further emphasises the importance of GAPDH for inflammation. Further characterisation of this post-translational modification is likely to advance our understanding of underlying processes in infection and inflammation, and potentially indicate new therapeutic strategies to limit inflammation in disease.

## Methods

**Reagents**. LPS was from Alexis. A/G plus agarose beads, streptavidin beads and biotin azide were obtained from Thermo Fisher. Sequencing grade modified trypsin was purchased from Promega. Heptelidic acid was bought from Abcam. Lipofectamine RNAiMax and Lipofectamine 2000 transfection reagents were obtained from Invitrogen. Malonyl-CoA, 2-deoxyglucose, anti-FLAG M2 affinity gel and 3X FLAG peptide were obtained from Sigma. Anti-malonyl-lysine (PTM-901), anti-GAPDH (MAB374), anti-GAPDH (ab8245), anti-IL1β (AF-401-NA), anti-ACC1 (4190) and anti-DAPK1 (3008) antibodies were obtained from PTM Biolabs, Merck, Abcam, R&D and Cell Signalling, respectively. Anti-β-actin (A3853), anti-myc (M4439), mouse IgG (5381) and rabbit IgG (15006) were obtained from Sigma. Peroxidase-conjugated anti-mouse, anti-rabbit and anti-goat secondary antibodies were purchased from Jackson Immunoresearch. Odyssey anti-mouse, anti-rabbit and anti-goat secondary antibodies were obtained from LiCOR Biosciences. HEK293T cell line was obtained from ATCC. All other reagents, unless otherwise especified, were obtained from Sigma.

**Mice and cell culture**. C57BL/6 mice were purchased from Harlam UK and maintained in GSK or Trinity Biomedical Science Insitute animal facilities under specific pathogen-free conditions. All animal studies were ethically reviewed and carried out in accordance with European Directive 86/609/EEC and the GSK Policy on the Care, Welfare and Treatment of animals.

Bone marrow cells were isolated from wild-type mice and differentiated in DMEM with 10% foetal bovine serum, 1% penicillin/streptomycin and 20% L929 supernatants for 6 days to generate BMDMs.

HEK293 cells were obtained from ATCC.

**Protein immunoprecipitation and western blotting**. A total of $10 \times 10^6$ cells/condition were lysed in low stringency buffer (50 mM HEPES pH 7.5, 100 mM NaCl, 1 mM EDTA, 10% glycerol, 0.5% NP40) with protease inhibitors. Protein concentration was measured using a BCA protein assay (Thermo Fisher) and normalised across samples prior to immunoprecipitation.

One millilitre of lysates were pre-cleared with 15 μL A/G beads for 30 min at 4 °C. For GAPDH immunoprecipitation, 3 μg of antibody pre-coupled to beads were added to lysates for 4 h at 4 °C. For mal-K immunoprecipitation, 9 μg of antibody and 50 μL of A/G beads were added overnight at 4 °C. Lysates were centrifuged for 3 min at 4 °C, the liquid was removed, and the beads were washed three times with low stringency buffer. Immune complexes were eluted by adding 30 μL of 5X Laemmli sample buffer and boiling for 5 min at 95 °C.

Protein samples from cultured cells were prepared by direct lysis of cells in 5X Laemmli sample buffer, followed by heating at 95 °C for 5 min. Protein samples were separated by SDS–PAGE and transferred to nitrocellulose or PVDF membranes via wet or iBlot (Invitrogen) transfer. Membranes were probed with the respective antibodies and visualised using LumiGLO enhanced chemiluminescent (ECL) substrate (Cell Signalling) or the Odyssey system. All primary antibodies were used in a 1:1000 dilution, with the exception of the anti-β-actin antibody, which was used in a 1:10,000 dilution. Secondary HRP-conjugated antibodies were used in a 1:2000 dilution and Odyssey secondary antibodies were used in a 1:10,000 dilution. The western blots for the enzyme assays were quantified using the Image Studio software from Odyssey, **also used** for obtaining the intensity ratios between GAPDH expression and β-actin.

Uncropped western blots from the main figures have been provided with the supplementary information (Supplementary Fig. 15).

**MalAMyne labelling and Cu(I)-catalysed click chemistry**. The MalAMyne chemical probe was synthesised as described by X. Bao et al.[27]. Cells were labelled with 10 μM MalAMyne or DMSO for 2 h. They were lysed and MalAMyne coupled to biotin through Cu(I)-catalysed click chemistry followed by streptavidin affinity enrichment as previously described[18].

**Malonylation mass spectrometry**. BMDMs were lysed in 8 M urea, 2 mM EDTA, 10 mM DTT, 1% Protease Inhibitor Cocktail, 2 μM TSA and 10 mM NAM, and cell debris removed by centrifugation at 20,000×g at 4 °C for 10 min. Proteins were precipitated with cold 15% TCA for 2 h at −20 °C, and then washed with cold

acetone three times. Proteins were dissolved in 8 M urea, 100 mM NH₄HCO₃, pH 8, and protein concentration measured using a BCA protein assay (Thermo Fisher). Protein concentration was normalised across samples.

Samples were reduced with 10 mM DTT, alkylated with 20 mM IAA and urea concentration reduced to 2 M using 100 mM NH₄HCO₃ followed by an overnight trypsin digestion. Tryptic peptides were then dissolved in NETN buffer (100 mM NaCl, 1 mM EDTA, 50 mM Tris–HCl, 0.5% NP-40, pH 8) and were incubated with pre-washed anti-malonyl-K antibody beads (PTM Biolabs) at 4 °C overnight. The beads were washed four times with NETN buffer and twice with ddH₂O. Bound peptides were eluted from the beads with 0.1% TFA, cleaned with C18 ZipTips (Millipore) following manufacturer's instructions, and analysed by LC–MS/MS.

Three parallel analyses for each fraction were performed. Peptides were dissolved in 0.1% formic acid and loaded onto a reversed-phase pre-column, Acclaim PepMap 100 (ThermoFisherScientific) and separated using a Acclaim PepMap RCLC (ThermoFisherScientific) at 700 nL/min in a gradient of 9–23% organic (0.1% FA in acetonrille) in 40 min, then followed by 12 min from 23% organic to 37% organic and finishing at 80% organic for 4 min. The samples were run on a Orbitrap Fusion (ThermoFisherScientific) coupled to a NanoEasy LC 1000. The resolution of the isntrument was set to 60,000 for MS, and data-dependent acquisition was performed, selecting the 20 most intense ions for MS/MS, the MS/MS resolution was set to 15,000.

The resulting MS/MS were processed using Maxquant search engine (v.1.5.2.8). Tandem mass spectra were searched against Swissprot *Mus musculus* database concatenated with reverse decoy database. Trypsin was specified as the cleavage enzyme allowing up to four missed cleavages. Main search range was set to 5 ppm and 0.02 Da for fragment ions. Carbamidomethyl on Cys was specified as fixed modification and malonyllysine on Lys and oxidation on Met were specified as variable modifications. Label-free quantification was performed using the Maxquant LFQ algorithm[57], by comparing the abundance of the same peptides across runs, with both ion intensities and spectral counts used for this purpose.

Mass spectrometry analysis was performed blindly by PTM Biolabs.

**GAPDH PTMs mass spectrometry**. For the identification of PTMs present in GAPDH, GAPDH was immunoprecipitated from 1 mg of BMDM lysates as previously described, and IP samples were separated on 4–12% SDS–PAGE gels, Coomasie stained, and bands of interest, together with their respective control bands, excised from the gel. Gel pieces were trypsin-digested and peptides dissolved in 1% TFA and injected onto a C18 spray tip and analysed by Q Exactive—Orbitrap mass spectrometer. The raw data was searched in Mascot. Trypsin was specified as the cleavage enzyme allowing up to one miscleavage and carbamidomethylation on cysteines specified as a fixed modification. Acetylation, succinylation, malonylation, ubiquitination and phosphorylation were specified as variable modifications.

Mass spectrometry analysis was performed blindly by PTM Biolabs.

**ELISAs and MSDs**. Cell culture supernatants were assayed for TNFα, IL6 and IL10 by ELISA (R&D) or by multiplex Meso Scale Discovery (MSD). MalonylCoA from lysates was assayed by ELISA (Cusabio).

**RNA extraction, reverse-transcriptase PCR and qPCR**. RNA was extracted using an RNeasy kit (Qiagen) and 250–1000 pg used for cDNA synthesis using a high capacity cDNA reverse transcription kit (Applied Biosystems). For qPCR, Taqman gene expression primers were used (Applied Biosystems), with expression of the target gene normalised to the geometrical mean of the expression of β-actin, 18S and GAPDH.

**siRNAs, mutagenesis and plasmids**. GAPDH siRNAs (s234321 and s103461), ACC1 siRNAs (s98860 and s98862) and ACSF3 siRNAs (s110944 and s110945) together with a silencer® select negative control (4390843), were obtained from Ambion. They were transfected into BMDMs seeded at $0.5 \times 10^6$ cells/mL at a concentration of 10 nM using Lipofectamine RNAiMax.

The following stealth siRNA duplex was used to KD human GAPDH:
Sense: CAUGUACCAUCAAUAAAGUACCCUG
Antisense: CAGGGUACUUUAUUGAUGGUACAUG
The following primers were used for mutagenesis reactions using the Quickchange II site-directed mutagenesis kit (Agilent Technologies):
GAPDH K213Q F: CTACTGGTGCTGCCCAGGCTGTGGGCAAGG
GAPDH K213Q R: GATGACCACGACGGGTCCGACACCCGTTCC
GAPDH K213E F: CTACTGGTGCTGCCGAGGCTGTGGGCAAGG
GAPDH K213E R: GATGACCACGACGGCTCCGACACCCGTTCC
The myc-DKK-GAPDH plasmid from Origene Technologies (Rockville, MD) (RC202309) was used as the template for mutagenesis and as the wild-type in experiments. psi-CHECK2 plasmid (Promega) was used for luciferase assays.

**Luciferase assay**. Five hundred nanograms of psiCHECK2, psiCHECK2-3'UTR-dapk1 and psiCHECK2-3'UTR-tnfa were transfected into HEK293T cells using Lipofectamine 2000. Forty-eight hours post-transfection cells were lysed with 1X passive lysis buffer (Promega). Forty microlitres of coelenterazine solution

(1 µg/mL, Promega) was added to 20 µL of lysate and read immediately on a luminometre to measure the renilla luciferase activity. Forty microlitres of 1X luciferase mix (2×: 20 mM Tricine; 2.67 mM $MgSO_4\cdot7H_2O$; 0.1 mM EDTA; 33.3 mM DTT; 530 µM ATP; 270 µM acetyl CoA; 470 µM D-Luciferin; 5 mM NaOH; 267 µM magnesium carbonate hydroxide) was added to 20 µL of lysate and the plate read immediately to measure the firefly luciferase activity.

**RNA-IP.** BMDMs ($10–20 \times 10^6$ cells/condition) were fixed with 1% formaldehyde and then neutralised with 1 M glycine. Cells were washed with PBS and lysed in polysome lysis buffer (0.1 M Hepes, 0.1 M KCl, 5 mM $MgCl_2$, 0.5% NP40, 1 mM DTT, 1X Protease Inhibitors). Five micrograms of GAPDH antibody or IgG control were pre-coupled to 50 µL of A/G beads and added to lysates. The samples were left rotating at 4 °C for 6 h. The beads were precipitated and washed at 4 °C five times with polysome lysis buffer.

After the last wash, the protein–RNA complexes were eluted with 85 µL of protein–RNA elution buffer (0.1 M Tris–HCl pH 8, 0.01 M EDTA, 1% SDS) and samples incubated at 37 °C twice. The eluates were then removed from the beads and 3 µL 5 M NaCl and 10 µg Proteinase K added. Samples were then incubated at 42 °C for 45 min to digest cross-linked peptides and then incubated for 1 h at 65 °C to reverse formaldehyde cross-links. The RNA was then extracted using phenol: chloroform extraction.

**GAPDH affinity purification and enzyme assay.** The chromatography columns were first rinsed with TBS (50 mM Tris–HCl, 150 mM NaCl, pH 7.4) twice, and the columns were then packed with 1 mL ANTI FLAG M2 affinity gel. The column was washed with 1 mL 0.1 M glycine HCl pH 3.5, three times, and then washed with 1 mL TBS five times.

A total of $10^7$ HEK293T cells overexpressing FLAG-GAPDH were lysed using 1 mL lysis buffer (50 mM Tris–HCl pH 7.4, 150 mM NaCl, 1 mM EDTA, 1%Triton) and 1X proteiase inhibitor. lysates were run through the column up to five times. The column was then washed 15 times with TBS. FLAG-GAPDH was eluted off the column by adding 5 mL of a 100 µg/mL 3X FLAG peptide solution.

GAPDH enymatic activity was assayed using a colorimetric GAPDH activity assay (Abcam).

**Polysome profiling.** Prior to lysis, cells were treated with cycloheximide (100 mg/mL), 10 min at 37 °C 5% $CO_2$. Cells were washed three times with ice cold PBS and lysed in ice cold buffer A (0.5% NP40, 20 mM Tris–HCl pH 7.5, 100 mM KCl and 10 mM $MgCl_2$). Lysates were passed three times through a 23 G needle and incubated on ice for 7 min. Extracts were then centrifuged at 10 krpm for 7 min at 4 °C. The supernatant was collected as crude cytosolic extract. Cytosolic extracts were overlaid on 10–50% sucrose gradients prepared in 20 mM Tris–HCl pH 7.5, 100 mM KCl and 10 mM $MgCl_2$ buffer (prepared using the Gradient Station, Biocomp Instruments). Gradients were then ultracentrifuged at 40 krpm for 1 h 20 min at 4 °C using an SW41 in a Beckman ultracentrifuge. Individual polyribosome fractions were subsequently purified using a Gradient Station (Biocomp Instruments). Total cellular RNA from BMDM cell lines or tissues was isolated using the Direct-zol™ RNA MiniPrep Kit (Zymo Research) according to manufacturer's instructions. RNA was quantified and controlled for purity with a nanodrop spectrometer. (Thermo Fisher). For RT-qPCR, 500–1000 ng were reversely transcribed (iScript Reverse Transcription Supermix, Biorad) followed by RT-PCR (iQ SYBRgreen Supermix, Biorad) using the cycling conditions as follows: 50 °C for 2 min, 95 °C for 2 min followed by 40 cycles of 95 °C for 15 s, 60 °C for 30 s and 72 °C for 45 s. The melting curve was graphically analysed to control for nonspecific amplification reactions. Quantitative RT-PCR analysis was performed with the following primers listed below:

Neat1 F: TTGGGACAGTGGACGTGTGG
Neat1 R: TCAAGTGCCAGCAGACAGCA
Gapdh F: CCAATGTGTCCGTCGTGGATC
Gapdh R: GTTGAAGTCGCAGGAGACAAC
Tnfα F: CAGTTCTATGGCCCAGACCCT
Tnfα R: CGGACTCCGCAAAGTCTAAG

**Statistical analysis.** Statistical tests were performed using GraphPad prism. A value of $p < 0.05$ was considered statistically significant.

**Reporting summary.** Further information on experimental design is available in the Nature Research Reporting Summary linked to this article.

## Data availability

Any further data not included in the manuscript is available from the corresponding author on reasonable request.

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

## Acknowledgements

We'd like to thank Steve Deharo for bioinformatic analysis input, as well as Mike Rees and Chris Mooney for mass-spectometry advice. We would also like to thank Emma Koppe for operational support within GSK. This work is supported by funding to the Luke O'Neill laboratory from the GlaxoSmithKline Visiting Science Programme and the Wellcome Trust (oneill-wellcometrust-metabolic, grant number 205455). This work also benefitted from data assembled by the ImmGen consortium.

## Author contributions

S.G.-P. designed and did experiments, analysed and interpreted data and wrote the manuscript; R.G.C. did experiments; A.N. synthesised the MalAMyne probe; A.M.J. performed 3D modelling; E.C.H. performed in vitro experiments; E.P.-M. performed the GAPDH Co-IP experiment; E.K.R., S.Co. and S.Ca. designed and performed polysome profiling; C.N. and M.H. helped design some experiments; M.P.M., L.K.M. and V.P.K. provided advice; L.A.O. funded and oversaw the research programme and edited the manuscript. All authors reviewed and approved the final version of the manuscript.

## Additional information

**Competing interests:** The authors declare no competing interests.

