## [Peer Review File · Nature Communications]

Reviewers' comments:

Reviewer #1 (Remarks to the Author):

Galvan-Pena et al. present interesting data suggesting that malonylation of GAPDH is induced after LPS stimulation in mouse macrophages. This malonylated GAPDH then can no longer bind TNF α mRNA and therefore may enhance translation of TNF α protein. This is novel and interesting, however I have two major concerns:

One point of concern is the relevance of the observed malonylation effects on GAPDH for TNF α production. The authors analyze malonylation of GAPDH and binding of GAPDH to TNF α mRNA at 0h and 24h. At 0h, there is neglectable TNF α mRNA levels present inside cells. At 24h, although there is still some TNF α mRNA present, TNF α is usually not produced at this late time point. So in order to corroborate the conclusions claimed in the manuscript, an earlier time point after LPS stimulation should be investigated (eg. 4, 6 or 8h). At this time point (although I recognize that no free access malonyl-CoA is present at 6h, there still should be an effect on GAPDH if the conclusions are valid), the following experiments should be performed:

- Malonylation levels of GAPDH
- Binding of GAPDH to TNF α mRNA

The second major point of concern is that the authors present no evidence that translation is indeed enhanced by malonylated GAPDH. A polysome profiling analysis or a similar experiment should be performed to support this claim.

Figure 3h shows accumulated protein levels over time; however, this does not show that at 24h there is still translation and secretion of TNF α as the authors suggest. Therefore, the stimulated cells should be washed at individual time points and TNF α levels in the supernatant can be determined by ELISA; or with intracellular cytokine staining.

Can the addition of Malonyl-Co to ACC1-KO macrophages increase expression of TNF α after LPS stimulation?

Is the level of malonylation of GAPDH reduced in ACC1 KO macrophages?

Can the HEK cells be stimulated with some stimuli that triggers TNF α production to substantiate the experiments with the mutant plasmids?

Minor:

Page 3: The authors mention that siRNA-mediated knock down of ACC1 or ACSF3 decreases protein levels by 50 % (Fig. 1c,d). This is not shown.

Fig. 1e: There are two LPS controls, which show different TNF α expression levels. If they are derived from different experiments, this should be indicated.

Fig. 1e,f: Time point of analysis is missing.

Fig. 3e: Maybe lower amounts of LPS would show a bigger increase of TNF α production in GAPDH KO cells?

Reviewer #2 (Remarks to the Author):

Pena et al described the malonylation of glyceraldehyde-3-phosphate dehydrogenase (GAPDH), a house keeping gene, regulating inflammatory signals in mouse macrophage cell-differentiated from bone marrow. This is an unique discovery in inflammatory immune response after LPS treatment. The data showed that a wide variety of proteins was malonylated by LPS stimulation that confirmed by mass spectrometer analysis. Authors found that the malonylated GAPDH bound to several mRNAs of inflammatory genes and particularly inhibited the translation of TNF α in resting macrophages. The inhibition of GAPDH activity on TNF α mRNAs was unleashed by LPS stimulation. LPS-mediated the modification of GAPDH malonylation dissociates GAPDH from TNF α mRNA that was happened after the K213 residue of GAPDH modified with upstream ACC1, but not ACC2. This eventually promotes TNF α translation. The results are well described by various experiments. These findings will contribute to the understanding of an unrevealed research field in inflammatory and immune response.

Comments:

1. Two isoforms of Acetyl-CoA carboxylase (ACC)-1 and -2 are expressed in mammalian tissues and cells that differ in distribution and function. It is known that the main structural difference between

two isoforms is the extended N-terminus containing a mitochondria targeting sequence in ACC2 isoform. ACC1 is found in the cytoplasm of all cells where fatty acid synthesis is important while oxidative tissues such as the skeletal muscle and the heart the ratio of ACC2 expressed is higher than ACC1.

Q: Is there any previous report or reference about the induction of ACC1 isoform by LPS as the result presented in figure 1? If not, please explain it in more detail to understand why these were occurred in your experiments.

2. Authors omitted legend for figure 4e.

Authors may need to add some description about figure 4e in legend as well as in main text, which is too short.

Please explain it more detail where LPS stimulates in figure 4e. Is it mitochondria where Krebs cycle occurred?

If this is a case, ACC2-associated mitochondria must be excluded with a sufficient explanation.

Reviewer #3 (Remarks to the Author):

In the manuscript "Malonylation of GAPDH is an inflammatory signal in macrophages", the authors report that GAPDH is malonylated during LPS treatment of macrophages. This process is suggested to be important for dissociation of GAPDH from TNF mRNAs and the efficient production of TNF protein. While the data presented on the identification of this posttranslational modification are compelling, the mechanistic and functional analysis are less impressive. Much of the data presented is correlative, with significant disconnects in the kinetics and functional activities described. These issues need to be corrected in a revised manuscript. Specific comments are listed below.

1. The data presented in figure 1, while intriguing, could use strengthening. Is it possible to knock out the genes encoding ACC1 or ASCF3? If not, then perhaps overexpression of these enzymes could be used to determine if their activity is rate limiting.

2. The data presented in figure 2f is surprising. Here, simply mixing GAPDH with malonyl-coA is sufficient to induce malonylation. One would expect that this posttranslational modification would be mediated by an enzyme, as all others are. The suggestion that GAPDH is its own malonyltransferase is a potential explanation, but more work would need to be done to establish this provocative conclusion. For example, can GAPDH induce malonylation of peptides that

correspond to K213? Is there a mutant of GAPDH that can be generated that abolishes this activity? These points are critical to address, as they are the foundation of the mechanistic model proposed.

3. The affects of LPS treatment on GAPDH activity and malonylation are shown at 24 hours post-treatment, yet it is recognized that glycolysis (and TNF translation) can occur within an hour of LPS exposure. Why does it take so long for GAPDH associated activities to occur?

4. Related to point 3, TNF secretion can be detected within a few hours of LPS treatment of macrophages, yet the affects of GAPDH on TNF production is only observed at 24 hours. One would expect that GAPDH needs to be removed from TNF transcripts immediately in order to allow translation and protein secretion. A detailed time course of GAPDH malonylation, GAPDH-TNF mRNA interactions and TNF translation needs to be performed.

Reviewers' comments:

Reviewer #1 (Remarks to the Author):

Galvan-Pena et al. present interesting data suggesting that malonylation of GAPDH is induced after LPS stimulation in mouse macrophages. This malonylated GAPDH then can no longer bind TNF α mRNA and therefore may enhance translation of TNF α protein. This is novel and interesting, however I have two major concerns:

1 - One point of concern is the relevance of the observed malonylation effects on GAPDH for TNF α production. The authors analyze malonylation of GAPDH and binding of GAPDH to TNF α mRNA at 0h and 24h. At 0h, there is neglectable TNF α mRNA levels present inside cells. At 24h, although there is still some TNF α mRNA present, TNF α is usually not produced at this late time point. So in order to corroborate the conclusions claimed in the manuscript, an earlier time point after LPS stimulation should be investigated (eg. 4, 6 or 8h). At this time point (although I recognize that no free access malonyl-CoA is present at 6h, there still should be an effect on GAPDH if the conclusions are valid), the following experiments should be performed:

- Malonylation levels of GAPDH
- Binding of GAPDH to TNF α mRNA

We have performed additional timepoint experiments as requested, which we agree help further strengthen our conclusions. We have measured malonylation levels of GAPDH at additional 6h and 12h LPS timepoints (Fig 2c). We found LPS gradually increases GAPDH malonylation over time with a small increase observed at 6h, which continues to increase at 12h. LPS-induced malonylation is observed to be at the highest at 24h. Additional earlier timepoints were tested (data not shown), and 6h was the earliest we could detect LPS-induced malonylation.

We also measured binding of GAPDH to TNF α mRNA at an additional 6h LPS timepoint (Fig 4a). We found the RNA-binding results nicely complemented that observed with GAPDH malonylation levels. A reduction in binding to the TNF α mRNA was observed at 6h, with no binding detected at 24h.

In addition, we have also provided an additional timepoint of TNF α measurement in the ACC1 KD BMDMs (Supplementary figure 2e, 2f), with similar levels of reduction observed both at 6h and 24h LPS.

2 - The second major point of concern is that the authors present no evidence that translation is indeed enhanced by malonylated GAPDH. A polysome profiling analysis or a similar experiment should be performed to support this claim.

We agree with this reviewer that the key to implicating translation in our model would be to perform a polysome profiling analysis. This is technically highly demanding, but we have been able to demonstrate that following LPS activation, there is increased association of TNF α mRNA with polysomes, indicating that there is increased translation (Supplementary figure 11). At the same time, we have also been able to identify via mass spectrometry that GAPDH is able to interact with polysome components at the time at which it is malonylated and releasing the TNF α mRNA, suggesting that GAPDH dissociation can transfer the RNA directly onto the polysomes to enable translation to occur (Fig 4d).

3 - Figure 3h shows accumulated protein levels over time; however, this does not show that at 24h there is still translation and secretion of TNF α as the authors suggest. Therefore, the stimulated cells should be washed at individual time points and TNF α levels in the supernatant can be determined by ELISA; or with intracellular cytokine staining.

We have performed an experiment in which BMDMs were treated with LPS, followed by removal of the LPS-containing media, followed by washing of the cells at the post-4h and post-8h LPS treatment. The media was replaced without any further LPS, and the supernatants harvested post 24h of the original LPS stimulus (20h and 16h post-washing, respectively). We were able to detect up to 1000 pg/mL of TNF α production by the cells via ELISA, indicating that the cells are actively translating and secreting TNF α at 24 hours. This data is shown in Fig 4c.

4 - Can the addition of Malonyl-Co to ACC1-KO macrophages increase expression of TNF α after LPS stimulation?

We set out to perform this experiment by first testing whether the cells would be able to take up malonyl-CoA and increase their intracellular concentrations of the metabolite. While the addition of mM concentrations of malonyl-CoA did not drastically increase intracellular concentrations of the metabolite (Supplementary fig 2f), there was enough of an observed increase to warrant testing of the model. We therefore proceeded to knockdown ACC1 in BMDMs and pre-treat the cells with malonyl-CoA prior to LPS treatment. The treatment with malonyl-CoA was able to recover the reduction in LPS-induced production of TNF α from the ACC1 KD, suggesting that malonyl-CoA production is indeed responsible for this effect (Fig 1h).

5 - Is the level of malonylation of GAPDH reduced in ACC1 KO macrophages?

Unfortunately, we do not have access to ACC1 knockout macrophages, as ACC1 KO mice are embryonically lethal¹. The system we have been using involves reducing ACC1 levels by siRNA-knockdown. To perform the suggested experiment would be very technically challenging, as we would need a very high number of primary cells to be able to include all the control groups (IgG control for IP, siRNA control, untreated vs LPS samples etc.) and have enough cells to immunoprecipitate enough GAPDH per sample to successfully detect malonylation levels. In addition, the siRNA transfection would have to be optimized for much bigger cell numbers than the ones we have been using, and the knockdown efficiency would have to be higher than 50% to be able to detect any malonylation changes visible by western blotting. Unfortunately, we have therefore not yet been able to perform this technically demanding experiment. Given however our other data supporting ACC1 as being needed for malonyl-CoA production and regulation of TNF α , and our multiple lines of evidence for GAPDH malonylation, we feel confident to conclude that ACC1 is needed for GAPDH malonylation.

6 - Can the HEK cells be stimulated with some stimuli that triggers TNF α production to substantiate the experiments with the mutant plasmids?

Unfortunately, there is no stimuli that we are aware of that would trigger TNF α production in HEKs. There is a transformed HEK cell line expressing TLR4/CD14 for LPS, but it is still not able to make TNF α as it is still missing signalling effectors.

Minor:

7 - Page 3: The authors mention that siRNA-mediated knock down of ACC1 or ACSF3 decreases protein levels by 50 % (Fig. 1c,d). This is not shown.

We apologize for this oversight, as we meant to state expression levels, as measured by qPCR. The text has been corrected. However, just to further validate the siRNA KD, we have now provided ACC1 protein expression results demonstrating knockdown at the protein level (Fig 1c). Unfortunately, we have been unable to successfully immunoblot for ACSF3.

8 - Fig. 1e: There are two LPS controls, which show different TNF α expression levels. If they are derived from different experiments, this should be indicated.

The controls for this figure panel are derived from the same experiment. One control well is set up for the ACC1 KD and a different control well is set up within the same plate for ACSF3 KD. It is usually the case that wells closer to the outer region of the plate suffer more evaporation than central ones, and as is the case here, it can result in variation from one control to the other. This variation is however non-significant.

9 - Fig. 1e,f: Time point of analysis is missing.

We apologize for this oversight and thank the reviewer for having spotted it. Time point of analysis has now been stated in the figure legend.

10 - Fig. 3e: Maybe lower amounts of LPS would show a bigger increase of TNF α production in GAPDH KO cells?

We have tested the suggested experiment and found while there was slightly non-significantly higher production of TNF α when using 10 ng/mL instead of 100 ng/mL of LPS, the increase in GAPDH knockdown cells was similar to that reported in Fig 3e.

Overall, we hope to have adequately addressed the concerns raised and thank the reviewer for their thorough review of this manuscript and for their comments, the addressing of which we feel have greatly strengthened the manuscript and provided further support for our model. We re-emphasize the multiple lines of evidence for GAPDH malonylation on K213 as a critical signal in macrophage activation by LPS. The evidence includes ACC1 being responsible for malonyl-CoA accumulation, GAPDH malonylation on K213 (multiple lines of evidence), decreased malonyl-CoA limiting TNF α production and GAPDH activity, and GAPDH repressing TNF α translation by binding its mRNA, with GAPDH dissociation via K213 malonylation relieving this repression to promote TNF α production.

Reviewer #2 (Remarks to the Author):

Pena et al described the malonylation of glyceraldehyde-3-phosphate dehydrogenase (GAPDH), a house keeping gene, regulating inflammatory signals in mouse macrophage cell-differentiated from bone marrow. This is an unique discovery in inflammatory immune response after LPS treatment. The data showed that a wide variety of proteins was malonylated by LPS stimulation that confirmed by mass spectrometer analysis. Authors found that the malonylated GAPDH bound to several mRNAs of inflammatory genes and particularly inhibited the translation of TNF α in resting macrophages. The inhibition of GAPDH activity on TNF α mRNAs was unleashed by LPS stimulation. LPS-mediated the modification of GAPDH malonylation dissociates GAPDH from TNF α mRNA that was happened after the K213 residue of GAPDH modified with upstream ACC1, but not ACC2. This eventually promotes TNF α translation. The results are well described by various experiments. These findings will contribute to the understanding of an unrevealed research field in inflammatory and immune response.

Comments:

1. Two isoforms of Acetyl-CoA carboxylase (ACC)-1 and -2 are expressed in mammalian tissues and cells that differ in distribution and function. It is known that the main structural difference between two isoforms is the extended N-terminus containing a mitochondria targeting sequence in ACC2 isoform. ACC1 is found in the cytoplasm of all cells where fatty acid synthesis is important while oxidative tissues such as the skeletal muscle and the heart the ratio of ACC2 expressed is higher than ACC1.

Q: Is there any previous report or reference about the induction of ACC1 isoform by LPS as the result presented in figure1? If not, please explain it in more detail to understand why these were occurred in your experiments.

We apologize if this wasn't made clear in the original manuscript submission. In figure 1b, we have measured expression of ACC1 in resting cells and not LPS-activated cells. We have also measured expression of all three enzymes in LPS-treated BMDMs (data not shown) and have not found LPS induces any change in the expression of any of the three. We have not been able to source any antibodies that can successfully detect ACC2 or ACSF3 via western blotting, but we have been able to detect protein expression of ACC1 (Fig 1c), and have not observed any LPS effects. In addition, we have also sought to validate our expression data with that of the publicly available RNAseq results from immune cells, and have found that the RNAseq data supports ACC1 as the highest expressed isoform, with no expression of ACC2 detected by RNAseq either (Supplementary figure 1). Interestingly, in doing so, we have stumbled upon the fact that macrophages express considerably much greater levels of ACC1 than any other immune cell type. In addition, while the bone marrow stem cell immune precursors do express ACC2, interestingly, most immune cells don't appear to express it. Other than the RNAseq data, there are only three studies that we have been able to find exploring the role of ACC1 in immune cells, which we have mentioned within the manuscript, but none of them measures the expression levels of any of the enzymes in macrophages.

2. Authors omitted legend for figure 4e.

Authors may need to add some description about figure 4e in legend as well as in main text, which is too short. Please explain it more detail where LPS stimulates in figure 4e. Is it mitochondria where Krebs cycle occurred? If this is a case, ACC2-associated mitochondria must be excluded with a sufficient explanation.

We thank this reviewer for spotting our oversight on the missing legend on figure 4e, which has now been

amended. In addition, we have significantly expanded the introduction and discussion text within the manuscript. We believe that ACC2 is not playing a role in this model, as per our explanation above, we can't find any evidence that ACC2 is expressed in macrophages. We believe that, based on previously published reports, LPS is likely having an effect on Krebs cycle first, causing a build-up of citrate, which can then be exported from the mitochondria and once converted into acetyl-CoA, stimulate ACC1 activity. It is also entirely possible, that LPS may be having an effect on ACC1 activity directly. The LPS effect on the ACC1 isoform, and the role of ACC1 in macrophages in general remains an area largely unexplored and which we believe holds great promise for future studies.

Overall, we hope to have adequately addressed the concerns raised and thank the reviewer for their review of this manuscript and for their comments, the addressing of which we feel have strengthened the manuscript and provided further support for our model.

Reviewer #3 (Remarks to the Author):

In the manuscript “Malonylation of GAPDH is an inflammatory signal in macrophages”, the authors report that GAPDH is malonylated during LPS treatment of macrophages. This process is suggested to be important for dissociation of GAPDH from TNF mRNAs and the efficient production of TNF protein. While the data presented on the identification of this posttranslational modification are compelling, the mechanistic and functional analysis are less impressive. Much of the data presented is correlative, with significant disconnects in the kinetics and functional activities described. These issues need to be corrected in a revised manuscript. Specific comments are listed below.

1. The data presented in figure 1, while intriguing, could use strengthening. Is it possible to knock out the genes encoding ACC1 or ASCF3? If not, then perhaps overexpression of these enzymes could be used to determine if their activity is rate limiting.

Unfortunately, while this is something that we'd definitely like to do, it is not yet technically possible. As far as we are aware, an ACSF3 KO mice has not been generated, and an ACC1 KO mice is embryonically lethal¹. There is a published study in which a macrophage specific ACC1 KO was generated by using ACC1^{fllox/fllox} mice crossed with CD11b-Cre, but they weren't able to achieve fully ACC1 KO either. They only observed 50% reduction in expression, which is similar to what we have obtained by using siRNAs. In addition, while we could consider transfecting CRISPR plasmids, macrophages are extremely difficult to transfect, as they are very good at sensing DNA, and transfections rates are usually at 10% if any at all. Unfortunately, the same applies for overexpression, as plasmids would also need to be used.

2. The data presented in figure 2f is surprising. Here, simply mixing GAPDH with malonyl-coA is sufficient to induce malonylation. One would expect that this posttranslational modification would be mediated by an enzyme, as all others are. The suggestion that GAPDH is its own malonyltransferase is a potential explanation, but more work would need to be done to establish this provocative conclusion. For example, can GAPDH induce malonylation of peptides that correspond to K213? Is there a mutant of GAPDH that can be generated that abolishes this activity? These points are critical to address, as they are the foundation of the mechanistic model proposed.

We'd like to clarify that we did not intend to rule out the possibility that malonylation could be mediated by an enzyme. We reported an interaction between GAPDH and the acyl-transferase p300, at the time of LPS-induced malonylation, so it is entirely possible that GAPDH malonylation may be mediated by p300. However, as we also observed rapid malonylation in vitro, we felt the possibility that GAPDH could be its own malonyl-transferase should also be considered. In fact, we believe both possibilities do not necessarily rule each other out, as in the case of acetylation, where both enzymatic and self-transfer have been reported, depending on different conditions.

We have attempted to perform further experiments to support the GAPDH malonyltransferase hypothesis, as suggested. To do so, we generated two different GAPDH active site mutants, C152S and C152A. However, we were unable to obtain consistent expression of the purified mutants. We believe that mutating the active site cysteine, may have resulted in disruption of the protein's structure, making the mutants unstable, and making any potential further assays using them inconclusive.

We currently do not have enough data to confirm a mechanism for malonylation, whether enzymatic or otherwise. The manuscript has been edited to make sure no overstatements are being made, in addition to providing additional discussion on the matter. As part of this manuscript, we report for the first time a signal

capable of triggering malonylation, as well as providing the first ever characterization of the modification in an immune cell population. We also identify via multiple lines of evidence, GAPDH activity as a key modulator of cytokine production in macrophages, and identify malonylation on K213 as the mechanism. In addition, we also for the first time report a role for ACC1 not in fatty acid metabolism, but as an enabler of malonylation via malonyl-CoA production. We believe that the question of how malonylation itself is occurring is an interesting one, and have therefore attempted to suggest potential possibilities via initial experiments. However, discovering the mechanism of a newly identified post-translational modification requires a significant amount of work that we believe lies beyond the scope of this paper and could constitute a standalone manuscript on its own, as demonstrated by prior acyltransferase and acylation mechanism publications²⁻⁴. We therefore hope that the reviewer will relent on this point.

3. The affects of LPS treatment on GAPDH activity and malonylation are shown at 24 hours post-treatment, yet it is recognized that glycolysis (and TNF translation) can occur within an hour of LPS exposure. Why does it take so long for GAPDH associated activities to occur?

We can detect GAPDH malonylation and GAPDH-associated activities as early as 6h of LPS treatment, but not earlier than that. We find that GAPDH malonylation and its activities certainly reach a peak at 24h post-treatment, and we believe that this is because of malonyl-CoA. While there is a study demonstrating increased glycolysis post 1h LPS in DCs⁵, within the same study citrate levels, which are needed for malonyl-CoA production, are reduced. In addition, we have previously reported an LPS timecourse on glycolysis in macrophages, and have not found an LPS-induced increase to occur any earlier than 6h⁶. There is also a detailed metabolic flux report in macrophages that reports citrate accumulation at 24h post-LPS treatment⁷. We are not aware of any other reports on macrophages demonstrating increased glycolysis post 1h LPS treatment. We believe that macrophages have mechanisms in place independently of GAPDH to kick start the early TNF α response. However at least 6h are needed for the macrophage to build up enough citrate to generate malonyl-CoA for LPS-induced malonylation to occur, enabling GAPDH to start an amplification wave and enable the cell to keep increasing the levels of TNF α production. We discuss this on page 27.

4. Related to point 3, TNF secretion can be detected within a few hours of LPS treatment of macrophages, yet the affects of GAPDH on TNF production is only observed at 24 hours. One would expect that GAPDH needs to be removed from TNF transcripts immediately in order to allow translation and protein secretion. A detailed time course of GAPDH malonylation, GAPDH-TNF mRNA interactions and TNF translation needs to be performed.

We have provided additional timepoints for both GAPDH malonylation (also measured now at 6h and 12h, fig 2c) as well as GAPDH RNA-binding (also measured now at 6h, fig 4a). This additional timepoints suggest that the malonylation effects are occurring gradually from 6h post-LPS onwards, ultimately peaking at 24h. We have also provided additional measurements of TNF α production having washed off the initial LPS stimulus post 4h and 8h, and demonstrated that the cells are actively translating and producing TNF α at later timepoints as far as 24h (Fig 4c). Finally we have been able to detect a marked increase in TNF α association with polysomes following LPS treatment via polysome profiling, at the same time as we have also been able to detect GAPDH associated with polysome components in response to LPS. While it is true that TNF α can be detected within a few hours of LPS treatment, there is very little TNF α protein being produced by the cells within the first 4 hours, compared to later timepoints, as depicted in fig 4b. We hope to have demonstrated that following the initial TNF α response to LPS, GAPDH is needed to amplify the TNF α response and enable it to continue to increase over time, with malonylation as the mechanism.

Overall, we hope to have adequately addressed the concerns raised and thank the reviewer for their review of this manuscript and for their comments, the addressing of which we feel have helped strengthened the manuscript and provided further support for our model. We re-emphasize the multiple lines of evidence for GAPDH malonylation on K213 as a critical signal in macrophage activation by LPS. The evidence includes ACC1 being responsible for malonyl-CoA accumulation, GAPDH malonylation on K213 (multiple lines of evidence), decreased malonyl-CoA limiting TNF α production and GAPDH activity, and GAPDH repressing TNF α translation by binding its mRNA, with GAPDH dissociation via K213 malonylation relieving this repression to promote TNF α production.

References

- 1 Abu-Elheiga, L. *et al.* Mutant mice lacking acetyl-CoA carboxylase 1 are embryonically lethal. *Proc Natl Acad Sci U S A* **102**, 12011-12016, doi:10.1073/pnas.0505714102 (2005).
- 2 Wang, Y. *et al.* KAT2A coupled with the alpha-KGDH complex acts as a histone H3 succinyltransferase. *Nature* **552**, 273-277, doi:10.1038/nature25003 (2017).
- 3 Du, J. *et al.* Sirt5 is a NAD-dependent protein lysine demalonylase and desuccinylase. *Science* **334**, 806-809, doi:10.1126/science.1207861 (2011).
- 4 James, A. M. *et al.* Non-enzymatic N-acetylation of Lysine Residues by AcetylCoA Often Occurs via a Proximal S-acetylated Thiol Intermediate Sensitive to Glyoxalase II. *Cell Rep* **18**, 2105-2112, doi:10.1016/j.celrep.2017.02.018 (2017).
- 5 Everts, B. *et al.* TLR-driven early glycolytic reprogramming via the kinases TBK1-IKKvarepsilon supports the anabolic demands of dendritic cell activation. *Nat Immunol* **15**, 323-332, doi:10.1038/ni.2833 (2014).
- 6 Tannahill, G. M. *et al.* Succinate is an inflammatory signal that induces IL-1beta through HIF-1alpha. *Nature* **496**, 238-242, doi:10.1038/nature11986 (2013).
- 7 Jha, A. K. *et al.* Network integration of parallel metabolic and transcriptional data reveals metabolic modules that regulate macrophage polarization. *Immunity* **42**, 419-430, doi:10.1016/j.immuni.2015.02.005 (2015).

REVIEWERS' COMMENTS:

Reviewer #1 (Remarks to the Author):

The authors have sufficiently addressed my concerns.

Reviewer #2 (Remarks to the Author):

I concerned about two issues in the manuscript. Authors answered the raised issues and addressed them properly in revised manuscript.

Reviewer #3 (Remarks to the Author):

In this revised manuscript, the authors have either experimentally addressed my concerns or explained why they could not address them. I remain curious about the precise enzymatic mechanisms underlying GAPDH malonylation, but agree that this point can be addressed in a future study. My questions of timing of malonylation as it relates to TNF translation were not addressed completely, but these concerns should not detract from the overall message of the study. I expect that this manuscript will be well-received by the community.